# Analytical Solution and Numerical Simulation of Heat Transfer in Cylindrical- and Spherical-Shaped Bodies



Humam Kareem Jalghaf [1,2] , Endre Kovács [1,*] , Imre Ferenc Barna [3] and László Mátyás [4]

1    Institute of Physics and Electrical Engineering, University of Miskolc, 3515 Miskolc, Hungary;
     20310@uotechnology.edu.iq
2    Department of Mechanical Engineering, University of Technology-Iraq, 19006 Baghdad, Iraq
3    Wigner Research Center for Physics, 1051 Budapest, Hungary; barna.imre@wigner.hu
4    Department of Bioengineering, Sapientia Hungarian University of Transylvania,
     530104 Miercurea Ciuc, Romania; matyaslaszlo@uni.sapientia.ro
*    Correspondence: kendre01@gmail.com or endre.kovacs@uni-miskolc.hu

**Abstract:** New analytical solutions of the heat conduction equation obtained by utilizing a self-similar Ansatz are presented in cylindrical and spherical coordinates. Then, these solutions are reproduced with high accuracy using recent explicit and unconditionally stable finite difference methods. After this, real experimental data from the literature regarding a heated cylinder are reproduced using the explicit numerical methods as well as using Finite Element Methods (FEM) ANSYS workbench. Convection and nonlinear radiation are also considered on the boundary of the cylinder. The verification results showed that the numerical methods have a high accuracy to deal with cylindrical and spherical bodies; also, the comparison of the temperatures for all approaches showed that the explicit methods are more accurate than the commercial software.

**Keywords:** numerical time integration; diffusion equation; heat equation; cylindrical coordinates; spherical coordinates; unconditional stability





## 1. Introduction

In nature and technology, transport processes are essential because they drive an important number of phenomena [1]. One of them is diffusion, where energy and particles are transported in a specific way [2,3]. The phenomenon of spreading in the universe occurs on a large scale from atoms to stars [4,5].

In the case of the models of two-dimensional diffusion, an important step has been realized by Machta and Zwanzig in Lorentz gas [6]. This work was followed by further simulations in Lorentz gas regarding the connection of diffusivity to certain phase space parameters [7] or to entropy [8]. Diffusion on surfaces is also a relevant topic. For cases where the surface may cause a chaotic dynamic, the phenomena was studied in Refs. [9–11].

Recently, exciting results have been obtained regarding diffusion with large spatial extension, which cover important analytical and computational results [12–15]. However, when the studied geometry of the system is either one-dimensional or two-dimensional, then the Cartesian coordinate system is used. In reality, the geometry of the studied system often drives the researcher to use other coordinate systems, most importantly spherical and cylindrical. A typical example is the case when a piece of ice is melting in water [16]. In nuclear physics, neutron diffusion occurs in finite spherical reactors [17]. Reactive diffusion through triple layers in spherical geometry was studied by Erdélyi and Schmitz [18] experimentally as well as numerically using the finite volume method. Cylindrical geometry was used to study diffusion in nanowires and nanorods by Roussel et al. [19]. The classic textbook [20,21] covers the theory of heat conduction in solids and that of diffusion and reaction in permeable catalysts, respectively, including spherically symmetrical cases. Furthermore, heat transfer in cylindrical coordinates finds numerous

applications, such as nuclear fuel rods [22,23], internal combustion engines [24], aluminum cylinders [25], and any bodies with curved surfaces, particularly curved surfaces in building walls. In these cases, finite difference discretization in polar or cylindrical coordinates offers significant advantages over Cartesian coordinates for solving boundary value problems involving circular shapes. Using polar or cylindrical coordinates eliminates the need for complex differentiation formulas near the curved boundaries, resulting in a more convenient and efficient computational process [26].

Heat conduction and similar problems are routinely solved using well-established numerical methods. However, in our view, these are far from being the optimal ones since they have serious drawbacks. Conventional explicit finite difference schemes are unstable when the applied time-step size exceeds the so-called Courant–Friedrichs–Lewy (CFL) limit, which is usually rather low. This is the main reason that implicit algorithms, which possess much better stability properties, are often used to tackle these equations [27–30]. These methods involve the solution of an algebraic equation system with the whole system matrix, which can be very large, especially if the number of spatial dimensions is larger than one. This is often time- and computer-memory consuming and cannot be straightforwardly parallelized.

In recent years, our research team developed some explicit methods that are stable for the heat conduction equation for arbitrary time-step sizes in any number of spatial dimensions (see [31] and the references therein). The algorithms were examined in several systems with homogeneous and inhomogeneous material parameters. They are orders of magnitude faster than the conventional solvers, including the standard Runge-Kutta schemes and the common MATLAB 'ode' solvers. However, all of these tests were performed in Cartesian coordinate systems, and now it is high time to perform them in cylindrical and spherical systems.

In the next section, we derive the studied equation and present its discretization. It is followed by the derivation of the analytical solution, which is valid in both cylindrical and spherical cases. This means that the isotropic heat equation is solved for two and three dimensions in case of infinite horizon. Section 4 is devoted to the description of the numerical algorithms, while Section 5 is about the verification of the methods using the analytical solution. Then, we turn our attention to examining the performance of the methods in reproducing experimental results: first, the circumstances are described, and then the numerical results of the Ansys software and our code are presented. Finally, the last section summarizes our conclusions.

## 2. The Studied Problem

We derived the heat transfer equation (conduction, convection, and radiation) in cylindrical and spherical coordinates based on energy balance. Firstly, in cylindrical coordinates, consider a small 3D cylindrical element $\Delta V = \Delta\phi(r + \frac{\Delta r}{2})\Delta r \times \Delta z$. In the case of full cylindrical symmetry, it is better to choose a full ring-shaped element, which yields $\Delta V = 2\pi(r + \frac{\Delta r}{2})\Delta r \times \Delta z = \pi\left((r + \Delta r)^2 - r^2\right)\Delta z$.

We assumed that physical quantities, including the temperature, did not change in the $\phi$-direction. This means we dealt only with two-dimensional problems from the mathematical point of view, as shown in Figure 1. The energy balance in this element during a time interval can be expressed as:

$$\begin{pmatrix} \text{Rate of heat conduction} \\ \text{at } r\,, z \end{pmatrix} - \begin{pmatrix} \text{Rate of heat conduction at} \\ r + \Delta r\,, z + \Delta z \end{pmatrix} + \begin{pmatrix} \text{Rate of heat generation inside} \\ \text{and on the surface of the element} \end{pmatrix}$$
$$\pm \begin{pmatrix} \text{Rate of convection} \\ \text{at the } r,\ z \text{ element} \end{pmatrix} \pm \begin{pmatrix} \text{Rate of radiation} \\ \text{at the } r,\ z \text{ element} \end{pmatrix} = \begin{pmatrix} \text{Rate of change of energy} \\ \text{content of the element} \end{pmatrix}$$

or, briefly,

$$Q_r + Q_\phi + Q_z - Q_{r+\Delta r} - Q_{\phi+\Delta\phi} - Q_{z+\Delta z} + Q_{gen} \pm Q_{convection} \pm Q_{radiation} = \frac{\Delta E_{element}}{\Delta t}. \tag{1}$$

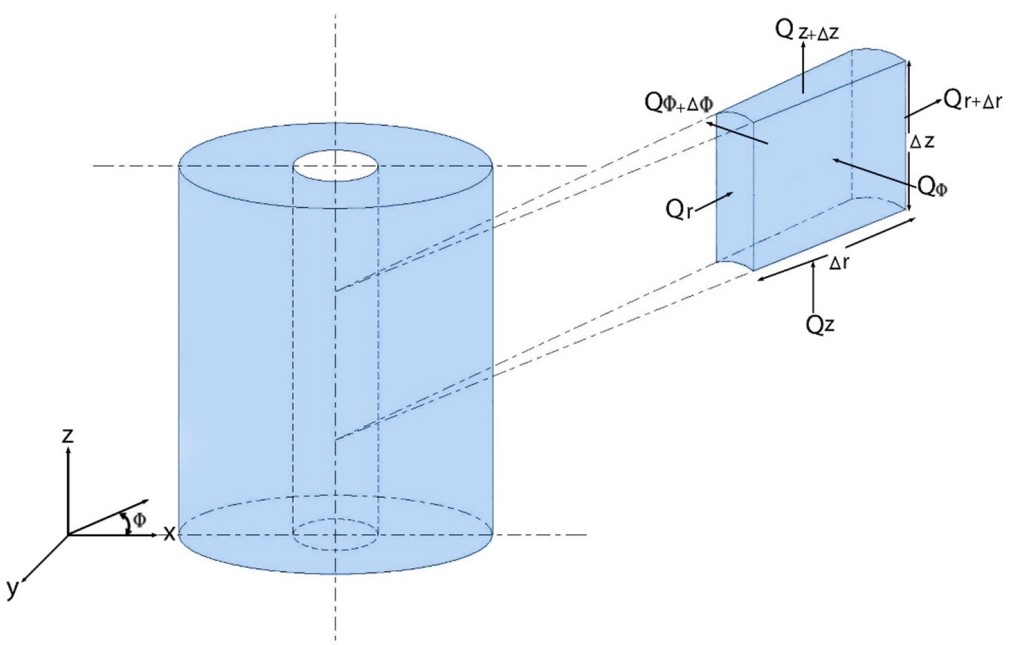

**Figure 1.** Visualization of a cylindrical element.

To fill Equation (1) with concrete formulas, the following three well-known laws were used.

Fourier's law of heat conduction:

$$Q_r = -kS\frac{\Delta u}{\Delta r}, \quad Q_z = -kS\frac{\Delta u}{\Delta z}, \tag{2}$$

In the given equation, $u$ represents the temperature, $k$ represents the thermal conductivity of the material, $S$ represents the surface area through which the heat flows, $r$ is the radius, and $t$ is the time.

Newton's law of heat convection can be stated as follows:

$$Q_{convection} = hS\Delta u = hS(u_a - u), \tag{3}$$

in the given equation, the symbol $h$ denotes the convection heat transfer coefficient. The ambient temperature $u_a$ is independent of the temperature $u$ of the body under examination. Hence, the term $hSu_a$ was included in the heat generation term to account for this effect.

The Stefan–Boltzmann law governs the incoming and outgoing heat radiation, and its expression is as follows:

$$Q_{radiation} = \sigma * S\left(u_a^4 - u^4\right), \tag{4}$$

In the given context $\sigma* = SB \cdot \varepsilon$, the universal Stefan-Boltzmann constant $SB = 5.67 \times 10^{-8}$ W/m$^2$K$^4$ needs to be multiplied by $\varepsilon = 85$, the emissivity constant, to account for the surface not being a black body. The incoming radiation $\sigma * Su_a^4$ is also considered in the heat source term $q$ as the $hSu_a$ absorptivity term. Consequently, the change in energy of an element over a specific time interval $\Delta t$ can be expressed as

$$\Delta E_{element} = E_{t+\Delta t} - E_t = mc(u_{t+\Delta t} - u_t) = \rho c\Delta V(u_{t+\Delta t} - u_t) \tag{5}$$

where $\rho = \rho(r)$ and $c = c(r)$ are the density and the specific heat, respectively.

From these equations, one can derive the heat-transport equation in a 3D cylindrical coordinate system, which can be written as:

$$\frac{1}{r}\frac{\partial}{\partial r}\left(kr\frac{\partial u}{\partial r}\right) + \frac{1}{r^2}\frac{\partial}{\partial \phi}\left(kr\frac{\partial u}{\partial \phi}\right) + \frac{\partial}{\partial z}\left(k\frac{\partial u}{\partial z}\right) + \frac{Q_{gen}}{\Delta V} - \frac{hS\,u}{\Delta V} - \frac{\sigma^*S\,u^4}{\Delta V} = \rho c\frac{\partial u}{\partial t} \tag{6}$$

In the case of spherical coordinates, a small 3D spherical element can be seen in Figure 2. The heat-transport equation for this case can be expressed as follows:

$$\frac{1}{r^2}\frac{\partial}{\partial r}\left(k\,r^2\frac{\partial u}{\partial r}\right) + \frac{1}{r^2\sin^2\theta}\frac{\partial}{\partial\phi}\left(k\,r\frac{\partial u}{\partial\phi}\right) + \frac{1}{r^2\sin\theta}\frac{\partial}{\partial\theta}\left(k\,\sin\theta\frac{\partial u}{\partial\theta}\right) + \frac{Q_{gen}}{\Delta V} - \frac{hS\,u}{\Delta V} - \frac{\sigma^* S\,u^4}{\Delta V} = \rho c\frac{\partial u}{\partial t}$$

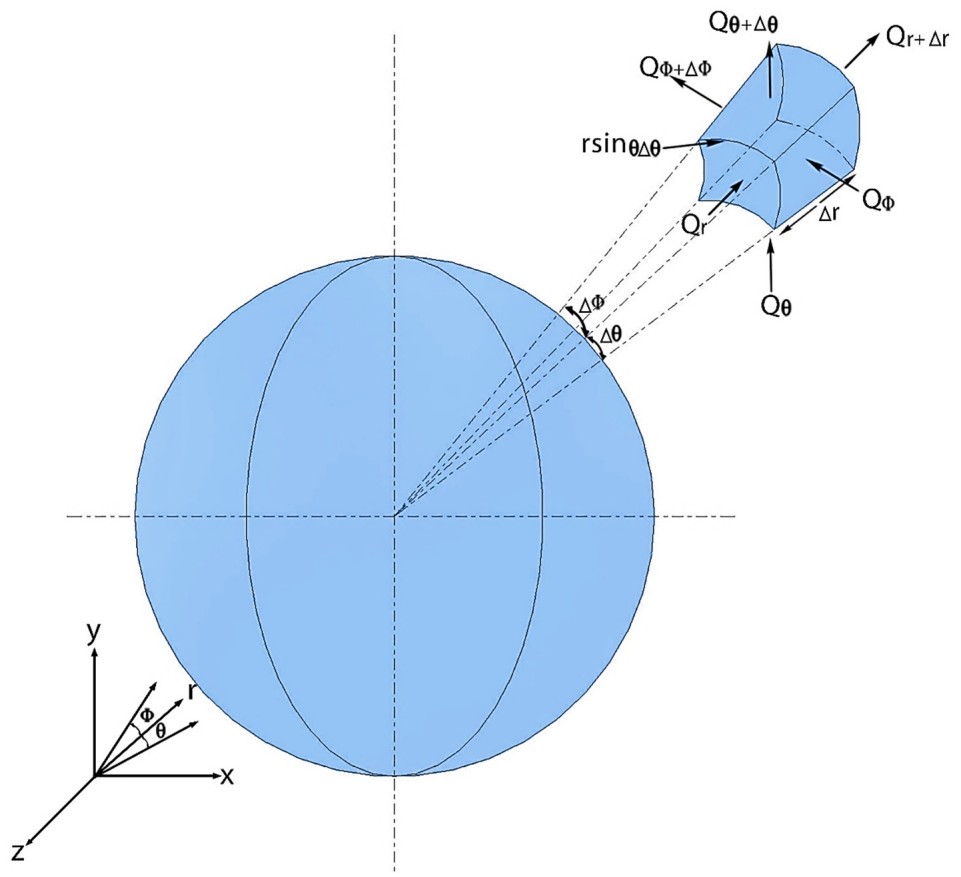

**Figure 2.** Visualization of a spherical element.

If one does not consider the convection, radiation, and source terms in Equation (6) and assumes that the material properties are homogeneous, one obtains the form of the heat conduction equation in cylindrical and spherical coordinate systems. We only investigated symmetrical systems, which means no relevant physical quantities depend on coordinate $\phi$ in the cylindrical and on coordinates $\phi$ and $\theta$ in the spherical case, which can be considered as a limitation of this study. If we—temporarily—also assume that nothing depends on the $z$ coordinate in the cylindrical case, only the radius $r$ remains as a spatial variable, which yields

$$\frac{\partial u}{\partial t} = \alpha\frac{1}{r^n}\frac{\partial}{\partial r}\left(r^n\frac{\partial u}{\partial r}\right), \tag{7}$$

where $n$ = 0, 1, and 2, which means Cartesian, cylindrical, and spherical coordinates, respectively, while $\alpha = \frac{k}{c\rho}$ is the (thermal) diffusivity. Equation (7) is also used for particle diffusion, where the diffusivity is usually denoted by $D$.

*The Spatial Discretization of the Problem*

Instead of directly discretizing PDEs (6) or (7), we use an equivalent resistance-capacitance model. In the case of cylindrical geometry, we consider tube-shaped cells with height $\Delta z$ and thickness $\Delta r$. For spheres, the cells have spherical-shell shapes with thickness $\Delta r$ again. The temperature is considered at the middle of the cell layer, where

the radial distance from the origin (the mean radius of the cells) is denoted by $r_i$, while the subsequent radius of the cell border is denoted by $r_i^* = r_i + \Delta r/2$.

The cell's heat capacity in the cylindrical and in the spherical case is approximated as $C_i = c_i \rho_i \pi (r_{i+1}^{*2} - r_i^{*2}) \Delta z$ and $C_i = c_i \rho_i \frac{4}{3} \pi (r_{i+1}^{*3} - r_i^{*3})$, respectively.

Let us denote the area of the cylindrical cell-surface perpendicular to $r$ with $S_r$, which can be given as $S_r = 2\pi r \Delta z$. Now, for the thermal resistance in the $r$-direction, the approximate formula

$$R_{i,i+1} \approx \int_{r_i}^{r_{i+1}} \frac{dr}{k_{i,i+1} S_r} = \int_{r_i}^{r_{i+1}} \frac{dr}{k_{i,i+1} 2\pi r \Delta z} = \frac{\ln(r_{i+1} - r_i)}{2\pi k_{i,i+1} \Delta z} \tag{8}$$

is used. For the thermal resistance in the $z$-direction, the approximate formula $R_{i,i+N_x} \approx \frac{(z_{i+N_r} - z_i)}{k_i \pi (r_{i+1}^2 - r_i^2)}$ is used, where the cell $i + N_r$ is below the cell $i$.

In the spherical case, $S_r$ can be given as $S_r = 4\pi r^2$. Using this, the thermal resistance is calculated similarly as that in the cylindrical case, but now the integration yields $R_{i,i+1} \approx \frac{1}{4\pi k_{i,i+1}} \frac{r_{i+1} - r_i}{r_i r_{i+1}}$. From Equations (2) and (5) it is easy to obtain the ODE system

$$\frac{du_i}{dt} = \sum_{j \neq i} \frac{u_j - u_i}{R_{i,j} C_i} + \frac{Q_{gen}}{C_i} - \frac{hSu_i}{C_i} - \frac{\sigma^* S u_i^4}{C_i} \tag{9}$$

to determine the time-evolution of the cell temperatures. Here, $S$ is the area of the surface on which the convection and radiation occurs, which will be the outer surface of the cylinder in Section 7. In publication [31], interested readers can find more details about this treatment of heat conduction. If one neglects the higher powers of $\Delta r$, one can easily derive that $C_i / S = c_i \rho_i \Delta r$ in both cases. Let us use the following notations:

$$K = \frac{h}{c\rho \Delta r} , \quad \sigma = \frac{\sigma^*}{c\rho \Delta r} , \quad q = \frac{\sigma^*}{c\rho \Delta r} u_a^4 + \frac{h}{c\rho \Delta r} \cdot u_a$$

Inserting these into (9), we can write Equation (9) in a simpler form:

$$\frac{du_i}{dt} = \sum_{j \neq i} \frac{u_j - u_i}{R_{i,j} C_i} + q_i - K u_i - \sigma u_i^4, \tag{10}$$

which will be solved numerically in Sections 5 and 7.

## 3. The Analytical Solution

In this section, we outline what the radial solutions of Equation (7) look like for both cylindrical and spherical coordinate systems. The self-similar Ansatz we used is as follows:

$$u(r, \ t) = t^{-a} f\left(r/\sqrt{t}\right) = t^{-a} f(\eta).$$

Substituting the first and second derivative of the Ansatz into the original Equation (2), we arrive at an ordinary differential equation (ODE) for $f(\eta)$

$$-af - \frac{\eta}{2} f' = \alpha \left( \frac{n f'}{\eta} + f'' \right), \tag{11}$$

with the constraint that $a$ is an arbitrary real number. The software Maple12 gives the following solution to this ODE:

$$f(\eta) \ = \ e^{\frac{-\eta^2}{4\alpha}} \left[ c_1 M\left( \frac{1}{2} + \frac{n}{2} + a, \ \frac{1}{2} + \frac{n}{2}, \ \frac{\eta^2}{4\alpha} \right) \ + c_2 U\left( \frac{1}{2} + \frac{n}{2} + a, \ \frac{1}{2} + \frac{n}{2}, \ \frac{\eta^2}{4\alpha} \right) \right], \tag{12}$$

Unfortunately, for unclear reasons, when substituting the $n = 0$ we cannot return to the original Cartesian solution; some factor is missing. Luckily, Equation (12) describes well the cylindrical and spherically symmetric solutions. Note that, unlike the Cartesian case, here all the solutions have the even property. In our former theoretical publications [12,32], we analyzed the possible different kinds of solutions. Depending on the numerical values of parameter *a*, four species of solution curves exist. This is generally true for Cartesian, cylindrical, and spherical systems as well. The only difference is just the numerical values of *a*.

Therefore, for large negative *a* values, the solutions show an exploding behavior at infinite temporal and spatial values, which obviously violates energy conservation and thermodynamics. When *a* is close to zero, we obtain solutions which have a finite non-zero numerical value at large times and special points. For small positive *a* values, the solution has a maximum at zero and has a similar decay to zero as the fundamental Gaussian solution. Finally, for large positive *a* values, the solutions have a power law explosion in the origin (at zero time and zero spatial coordinate) with a drastic decay accompanied by additional oscillations. Therefore, the solutions become negative at some $\eta$ values. These are the general new features of these kinds of solutions for Cartesian, cylindrical, and spherical systems as well. Figure 3 presents the Kummer's M functions as solutions for three different parameter sets. The left sub-figure is for the cylindrical and the right sub-figure is for the spherically symmetric case.

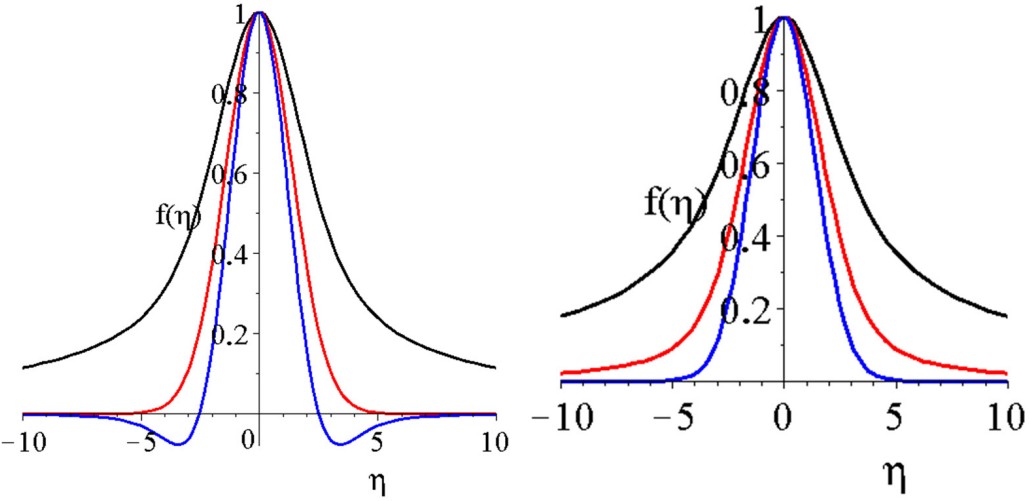

**Figure 3.** The Kummer's M shape functions f($\eta$) from Equation (9) for the cylindrical (**left**) and spherical (**right**) symmetry for three different self-similar exponents *a*. The black, red, and blue lines are for $a = 1/2$, 1, and $3/2$. The diffusion coefficient $\alpha$ was set to unity for all cases.

For our forthcoming numerical analysis, instead of solution (12) of the ODE (11), we need the final form of the solution of PDE (7), which reads as follows:

$$u^{\text{exact}}(r,\ t) = t^{-a} e^{-\frac{r^2}{4\alpha t}} \left( c_1\, M\left(\frac{1}{2} + \frac{n}{2} + a,\ \frac{1}{2} + \frac{n}{2},\ \frac{r^2}{4\alpha t}\right) + c_2\, U\left(\frac{1}{2} + \frac{n}{2} + a,\ \frac{1}{2} + \frac{n}{2},\ \frac{r^2}{4\alpha t}\right) \right), \quad (13)$$

Figure 4 presents the $u(r,\ t)$ solution for Kummer's M function for cylindrical symmetry. One can observe certain similarities with the regular Gaussian.

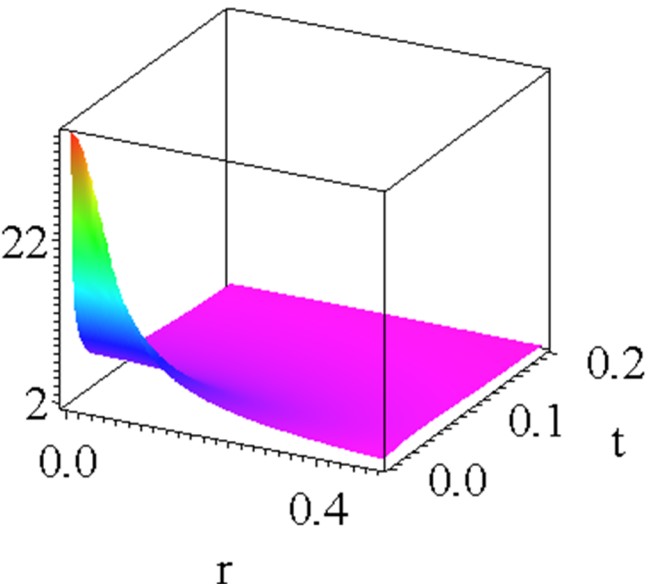

**Figure 4.** The Kummer's M part of the solution $u(r, t)$ for a = 1/2 self-similar exponent and $\alpha = 1$ diffusion coefficient for $n = 1$, cylindrical symmetry.

### 4. The Applied Numerical Methods

We present the numerical schemes adapted for Equation (10). We always used equidistant discretization of the time variable with time-step size $\Delta t$, and the temperature of cell $I$ at the time $n\Delta t$ is denoted by $u_i^n$. The following quantities were extensively used:

$$w_i = \Delta t \sum_{j \neq i} \frac{1}{C_i R_{ij}} \text{ and } A_i = \Delta t \sum_{j \neq i} \frac{u_j^n}{C_i R_{ij}} + \Delta t \cdot q_i.$$

The quantity $w$ is the generalization of the so-called mesh-ratio (which has the expression $\frac{\alpha \Delta t}{\Delta x^2}$), but it now depends on the cell index. The aggregated quantity $A$, on the other hand, is responsible for the conductive heat exchange with the neighbors of cell $i$ as well as the source term.

1.  The UPFD (unconditionally positive finite difference) method was proposed in [33] for the diffusion-advection-reaction equation a decade ago. We recently adapted it to Equation (10) as follows:

$$u_i^{n+1} = \frac{u_i^n + A_i}{1 + w_i + \Delta t \cdot K_i + \Delta t \cdot \sigma_i \cdot (u_i^n)^3}.$$

2.  The pseudo-implicit (PI) method has the following two stages:

$$Stage 1: u_i^{pred} = \frac{u_i^n + A_i/2}{1 + w_i + \Delta t \cdot K_i/2 + \Delta t \cdot \sigma_i \cdot (u_i^n)^3/2},$$

$$Stage 2: u_i^{n+1} = \frac{(1 - w_i)u_i^n + A_i^{new} + \Delta t \cdot K_i \left( u_i^{pred} - u_i^n \right)}{1 + w_i + \Delta t \cdot K_i + \Delta t \cdot \sigma_i \cdot (u_i^n)^3}, \text{ where } A_i^{new} = \Delta t \sum_{j \neq i} \frac{u_j^{pred}}{C_i R_{ij}} + \Delta t \, q_i$$

3.  A two-stage version of the Rational Runge–Kutta methods [34] was applied as follows. First, a full step was taken by the standard FTCS (explicit Euler) scheme to calculate the predictor values:

$$u_i^{pred} = u_i^n + g_i^1,$$

where the increment $g_i^1$ is calculated as

$$g_i^1 = A_i - \Delta t \cdot K_i \cdot u_i^n - \Delta t \cdot \sigma_i \cdot (u_i^n)^4.$$

Using these predictor values, the increment in the second stage was

$$g_i^2 = A_i^{\text{new}} - \Delta t \cdot K \cdot u_i^{\text{pred}} - \Delta t \cdot \sigma \cdot (u_i^{\text{pred}})^4.$$

Now, the new values of the variable are as follows

$$u_i^{n+1} = u_i^n + \frac{2p_1 g_i^1 - 2p_{12} g_i^1 + p_1 g_i^2}{4p_1 - 4p_{12} + p_2}.$$

where the following scalar products were used:

$$p_1 = \left(\vec{g}^1, \vec{g}^1\right) = \sum_{i=1}^{N} g_i^1 g_i^1, \quad p_{12} = \left(\vec{g}^1, \vec{g}^2\right) = \sum_{i=1}^{N} g_i^1 g_i^2, \quad p_2 = \left(\vec{g}^2, \vec{g}^2\right) = \sum_{i=1}^{N} g_i^2 g_i^2.$$

4. A special, checkerboard-like spatial grid must be constructed if someone wants to use any version of the odd-even hopscotch methods [35]. The cells of this grid are labeled as odd and even, with the requirement that all the nearest neighbors of the even cells are odd and vice versa. In the case of the original version (denoted by OOEH here), the odd-even labels must be interchanged after each time step, as is displayed in Figure 5A. The standard FTCS formula was modified in the first stage to make it slightly more stable [31] by treating the convection and radiation terms in an "implicit" way:

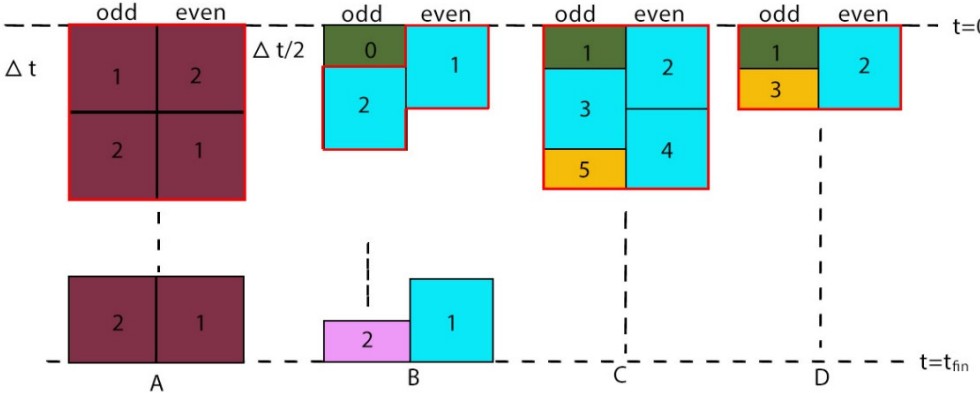

**Figure 5.** Space-time structure of (**A**) the original hopscotch methods. (**B**) The leapfrog-hopscotch method. (**C**) The shifted-hopscotch method. (**D**) The asymmetric hopscotch method [31].

$$u_i^{n+1} = \frac{(1 - w_i)u_i^n + A_i}{1 + \Delta t \cdot K_i + \Delta t \cdot \sigma_i \cdot (u_i^n)^3}$$

The implicit (Euler) scheme in the second stage was as follows:

$$u_i^{n+1} = \frac{u_i^n + A_i^{\text{new}}}{1 + w_i + \Delta t \cdot K_i + \Delta t \cdot \sigma_i \cdot (u_i^n)^3}$$

It is important that in the second stage, similarly to the other odd-even hopscotch scheme below, that the latest-available values of the neighbors are always used when the new values of $u$ are calculated. Due to this, the $u_j^{n+1}$ values in the implicit formula were just obtained in Stage 1, which makes it effectively explicit.

5. In the case of the shifted-hopscotch (SH) algorithm, five stages constitute a two-step-long repeating block. Two of the stages are half- and the remaining three of them are full-length steps, as seen in Figure 5C. The first stage is for the odd cells only and is symbolized by a dark green box with the number one in the figure. It uses the formula

$$u_i^{n+1/2} = \frac{u_i^n + A_i}{1 + w_i + \Delta t \cdot K_i + \Delta t \cdot \sigma_i \cdot \left(u_i^n\right)^3}. \tag{14}$$

Then, a full time-step with formula

$$u_i^{\mu+1} = \frac{(1 - w_i/2)u_i^\mu + A_i^{\mu+1/2}}{1 + w_i/2 + \Delta t \cdot K_i + \Delta t \cdot \sigma_i \cdot \left(u_i^\mu\right)^3} \tag{15}$$

for the even, the odd, and the even cells come again (blue rectangles in the figure). The superscript $\mu$ equals $n$ for the even and $n + 1$ for the odd cells. Finally, a half-length time step (pink box with number five inside) for the odd cells closes the calculation with the formula

$$u_i^{n+2} = \frac{(1 - w_i)u_i^{n+1} + A_i^{n+2/3}}{1 + \Delta t \cdot K + \Delta t \cdot \sigma \cdot \left(u_i^{n+1}\right)^3}. \tag{16}$$

6. The asymmetric hopscotch (ASH) algorithm is almost the same as the SH one, but the repeating block is only one time-step long, as one can see in Figure 5D. It contains only three stages instead of five, which use the same Formulas (14)–(16), respectively.
7. The procedure of the leapfrog-hopscotch (LH) algorithm begins and ends with a half-length stage, but then all other stages have full time-step length (blue boxes in Figure 5B). At the first and intermediate stages, it uses Formulas (14) and (15). However, at the last stage (pink rectangle) it uses (15) again, but $\Delta t$ must be divided by two, including their appearances in the quantities $w$ and $A$.
8. Although the Dufort–Frankel (DF) method [36] is a known explicit and unconditionally stable algorithm for the linear heat equation, it is rarely used. The formula adapted for the case of Equation (10) is:

$$u_i^{n+1} = \frac{(1 - w_i)u_i^{n-1} + 2A_i}{1 + w_i + 2 \cdot \Delta t \cdot K_i - 2 \cdot \Delta t \cdot \sigma_i \cdot \left(u_i^n\right)^3}.$$

It is a two-step scheme which is not self-starter. Thus, we applied the UPFD formula of point 1 to start the DF method.

9. One of the most common algorithms to solve the heat conduction equation is the explicit-Euler-based FTCS (forward time central space) algorithm. It can be adapted to our case in the standard way as follows:

$$u_i^{n+1} = (1 - w_i)u_i^n + A_i - \Delta t \cdot K_i \cdot u_i^n - \Delta t \cdot \sigma_i \cdot \left(u_i^n\right)^4$$

## 5. Verification Using the Analytical Solution

In this section, we take the height of the cylinder as well as $\Delta z$ unity. It means that, computationally, there is one space dimension only in both the cylindrical and the spherical case. The solution parameters are:

$N_r = 500$, $N_z = 1$, $N = N_r \times N_z = 500$, $r_0 = 0.0003$, $r_{max} = 0.999$, $\Delta r = 0.002$,
$\alpha = 1$, $a \in \{1, 1.2, 2\}$, $c_1 = 1$, $c_2 = 0$, $t^0 = 0.1$, $t^{fin} = t^0 + 0.1$.

Here, $N$ represents the total number of cells, while $r_0$ and $r_{max}$ are the radial coordinates of the center of the first and last cells. The CFL limit (maximum allowed time-step

size for the standard first order forward Euler method) was around $2 \cdot 10^{-6}$ in all cases. The initial condition was obtained by substituting the initial $t$ and boundary $r$ values into the analytical solution, respectively. The Dirichlet boundary conditions on the right side (the circumference of the cylinder and sphere) were obtained simply by substituting the radius $r_{\max}$ into the analytical solution and calculating the function value at each time step. On the left side (the cylinder and sphere center, $r = r_0$), zero-Neumann boundary was applied, since no heat can disappear from the center of the cylinder or the sphere. This boundary was applied only computationally and not physically. We remind the reader that the analytical solutions are constructed for Equation (7), and here we accurately reproduce them by solving the apparently different Equation (10) numerically.

We calculated the so-called maximum error:

$$MaxError = \max_{1 \leq i \leq N} \left| u_i^{\text{analytic}}(t_{\text{fin}}) - u_i^{\text{num}}(t_{\text{fin}}) \right|, \tag{17}$$

The obtained maximum errors are displayed as a function of the time-step size in Figures 6 and 7 for two values of parameter *a* in cylindrical coordinates. Figure 8 presents the temperature value as a function of *r*. For the case of spherical coordinates, Figure 9 shows the maximum error as a function of the time step, and Figure 10 presents the temperature as a function of *r*. The fact that we obtained very small errors in all cases verifies not only the numerical algorithms, but the equivalence of the two mathematical treatments of the physical problem.

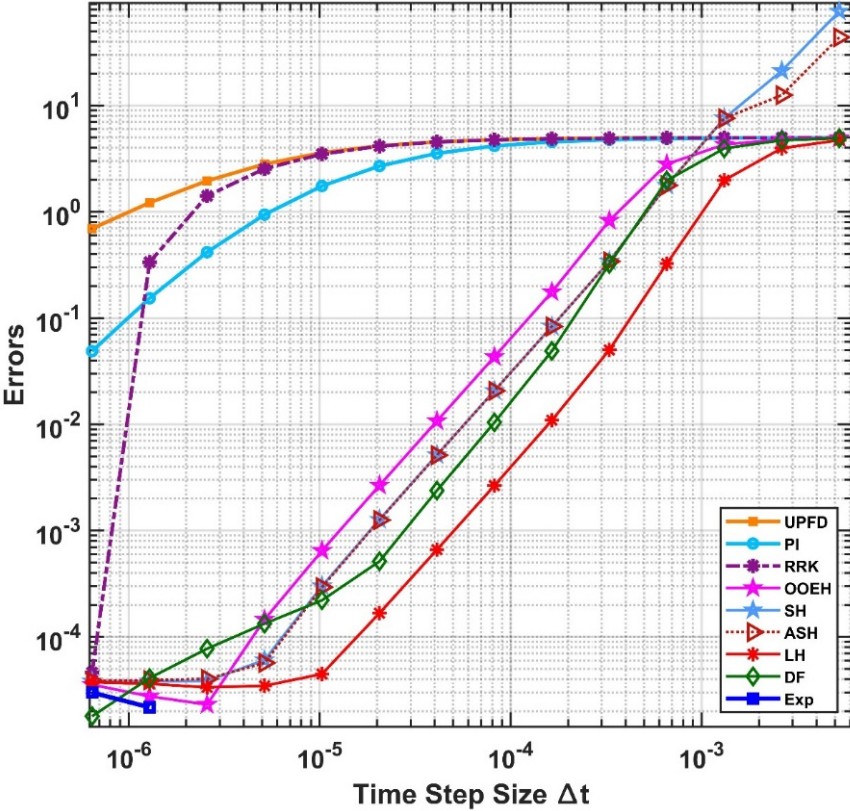

**Figure 6.** The maximum errors as a function of the time step size $\Delta t$ for the 9 numerical methods in case of cylindrical coordinates for *a* = 1.

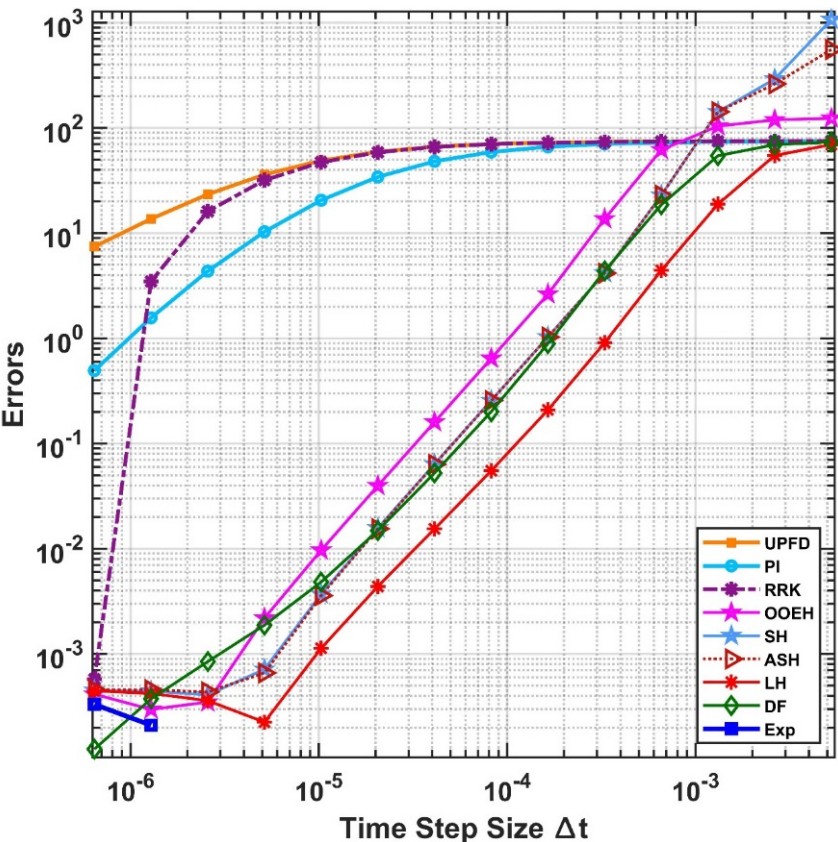

**Figure 7.** The maximum errors as a function of the time step size for the 9 numerical methods in case of cylindrical coordinates for $a = 2$.

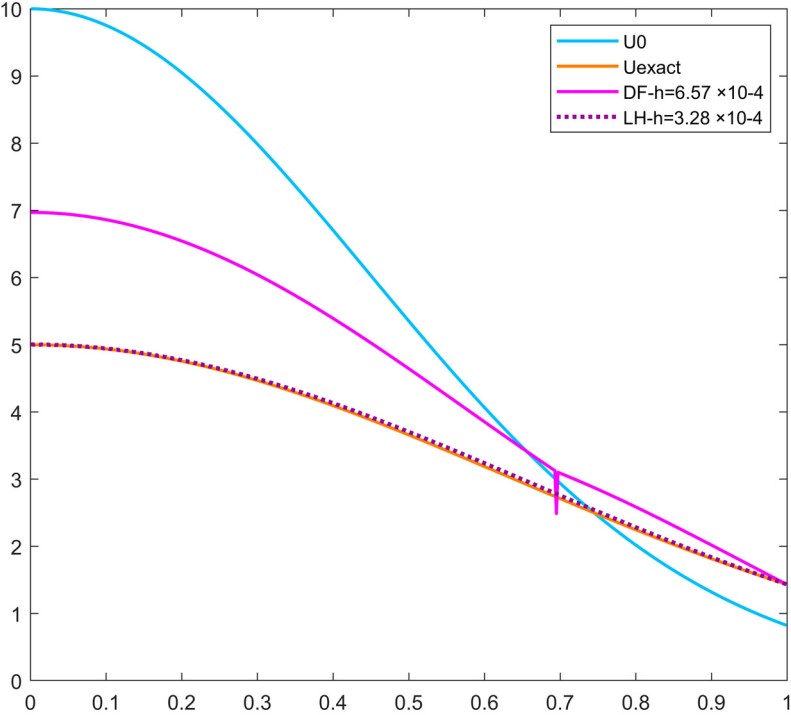

**Figure 8.** The values of temperature as a function of variable $r$ in case of the initial function $u^0$, the analytical solution Uexact, the DF method, and the LH method in case of cylindrical coordinates for $a = 1$.

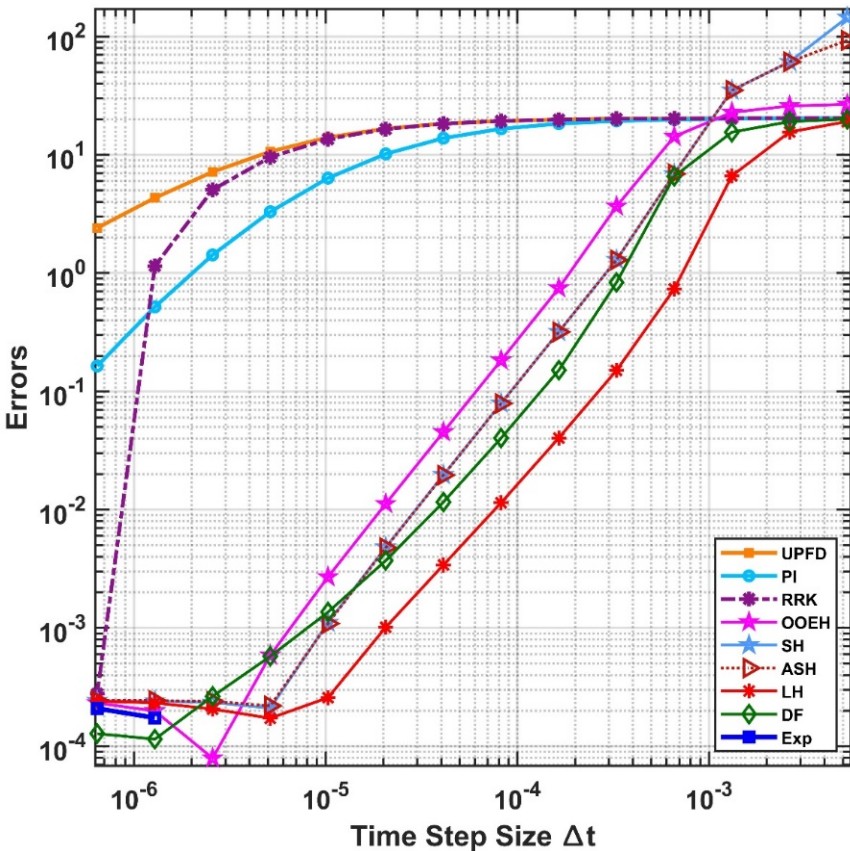

**Figure 9.** The maximum errors as a function of the time step size for the 9 numerical methods in case of spherical coordinates for $a = 1.2$.

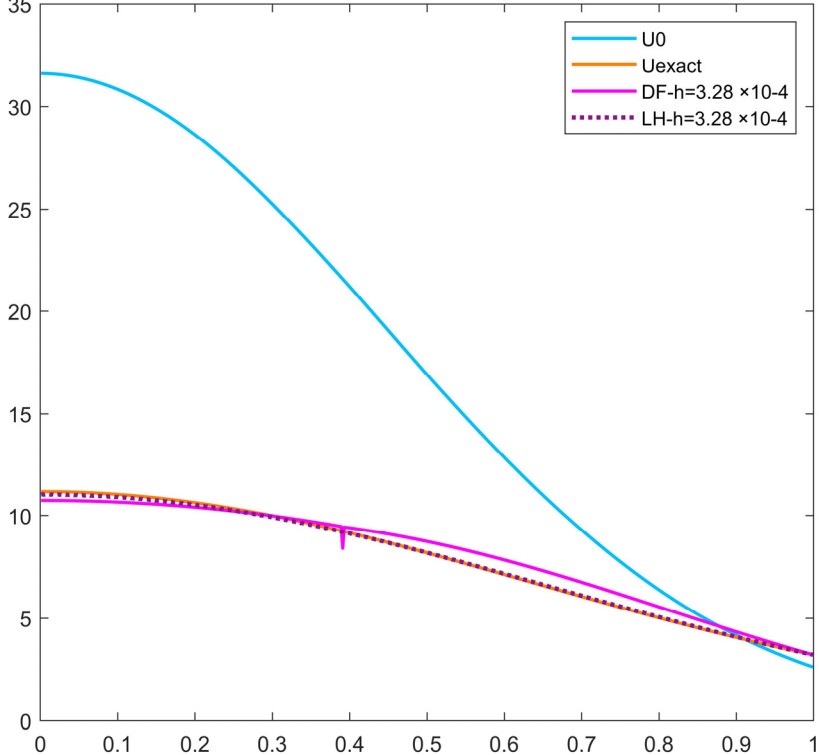

**Figure 10.** The values of temperature as a function of $r$ variable in case of the initial function $u^0$, the analytical solution Uexact, the DF method, and the LH method in case of spherical coordinates for $a = 1.2$.

## 6. Setup of the Reproduction of the Experimental Results

### 6.1. Material Properties

We are going to reproduce the experimental results of Cabezas et al. [37], where heat transfer was studied in a steel C45 cylinder of 168 mm total height with properties shown in Table 1 below.

**Table 1.** The properties of the steel used [38].

| Material | $\rho$ $(\mathrm{kg \cdot m^{-3}})$ | $k\left(\mathrm{W \cdot m^{-1} \cdot K^{-1}}\right)$ | $c\left(\mathrm{J \cdot kg^{-1} \cdot K^{-1}}\right)$ |
|:---:|:---:|:---:|:---:|
| Steel C45 | 7800 | 40 | 480 |

The bottom of the cylinder was heated for 30 s at the beginning of the experiment with $P$ = 1500 W power. However, in the original work [37], the position of the lowest thermocouple was 50 mm higher than the heated surface. The top 118 mm and not the bottom 50 mm of the cylinder was examined either experimentally or numerically, and we followed this in our work. This means that the simulated volume of the cylinder segment is $V = 1.0087 \times 10^{-4}$ m$^3$, while $(r, z) \in [0, \ 0.0165$ m$] \times [0, \ 0.118$ m$]$. In our approximation, physical quantities did not change in the $\phi$-direction. Thus, that dimension was irrelevant and, computationally, we dealt with a two-dimensional problem. The number of the cells along the $r$ axis and $z$ axis were set to $N_\mathrm{r}$ = 15 and $N_\mathrm{z}$ = 100; thus, the total number of the cells in the system was $N = N_r N_z = 1500$. We emphasize that we used the same code as in Section 5; only some parameters, such as $N_\mathrm{z}$, are different.

### 6.2. The Initial and the Boundary Conditions

We use a constant initial condition in all cases.

$$u(r, z, \ t = 0) = 30.7 \, ^\circ \mathrm{C}$$

As in Section 4, we used different boundary conditions on different sides. On the left side, the center of the cylinder, we applied Neumann boundary conditions in all cases, which do not allow conductive heat transfer at the boundary,

$$u_r(r = 0, z, t) = u_r(r = L_r, z, t) = u_z(r, z = L_z, t) = 0 \tag{18}$$

On the right (external) and upper boundaries, we used two types of boundary conditions. The first one was zero-Neumann, when there was no heat exchange with the environment. The second one, when there was a heat exchange with the environment via convection and radiation, considered the heat convection coefficient $h = 4.5\left(\mathrm{W \cdot m^{-2} \cdot K^{-1}}\right)$ [38] and the emissivity constant as 0.85 to obtain realistic values for $\sigma*$. The convective and radiative energy transfer was perpendicular to the surface. The interior elements cannot gain or lose heat by the heat source, heat convection, or radiation.

On the lower boundary, we applied changing Dirichlet boundary conditions based on the temperature measurement results taken from a report we asked from the authors of [37]. That report contained data from every two minutes, and we used linear interpolation between these data points in all cases to follow the experimental setup of the paper [37].

The heat generation contained incoming heat via convection and radiation depending on the ambient temperature. Since the steel cylinder was placed in a closed box, this ambient temperature changed during the measurement. Instead of the ambient temperature functions, we used their averages taken from the report mentioned above. The ambient temperature of the air was taken as (30.7, 31.1, and 31.7 °C) in the cases of measurements at 20 min, 24 min, and 30 min duration, respectively.

## 7. Simulation Results

In this section, the end of the examined time interval is defined as $t_{\text{fin}} = 1200,\ 1440$ and $1800$ s. To ensure that the errors of the numerical algorithms were small, we used a reference solution of the spatially discretized Equation (9) obtained using the ode15 s solver in MATLAB with a small tolerance $10^{-10}$. This reference solution was utilized in Equation (17) to determine the maximum error of the time integration, which did not contain the error of the mathematical model and that of the space discretization.

### 7.1. Results of the Numerical Methods

For the simulation, we chose the top five algorithms, namely DF, OOEH, LH, SH, and ASH. The simulation of a steel C45 cylinder was conducted using these selected algorithms considering different boundary conditions, as previously mentioned. Among these algorithms, the shifted-hopscotch method was chosen to visualize the temperature contour due to its high accuracy at small time-step size. Figures 11 and 12 display the final temperature distribution obtained from this method.

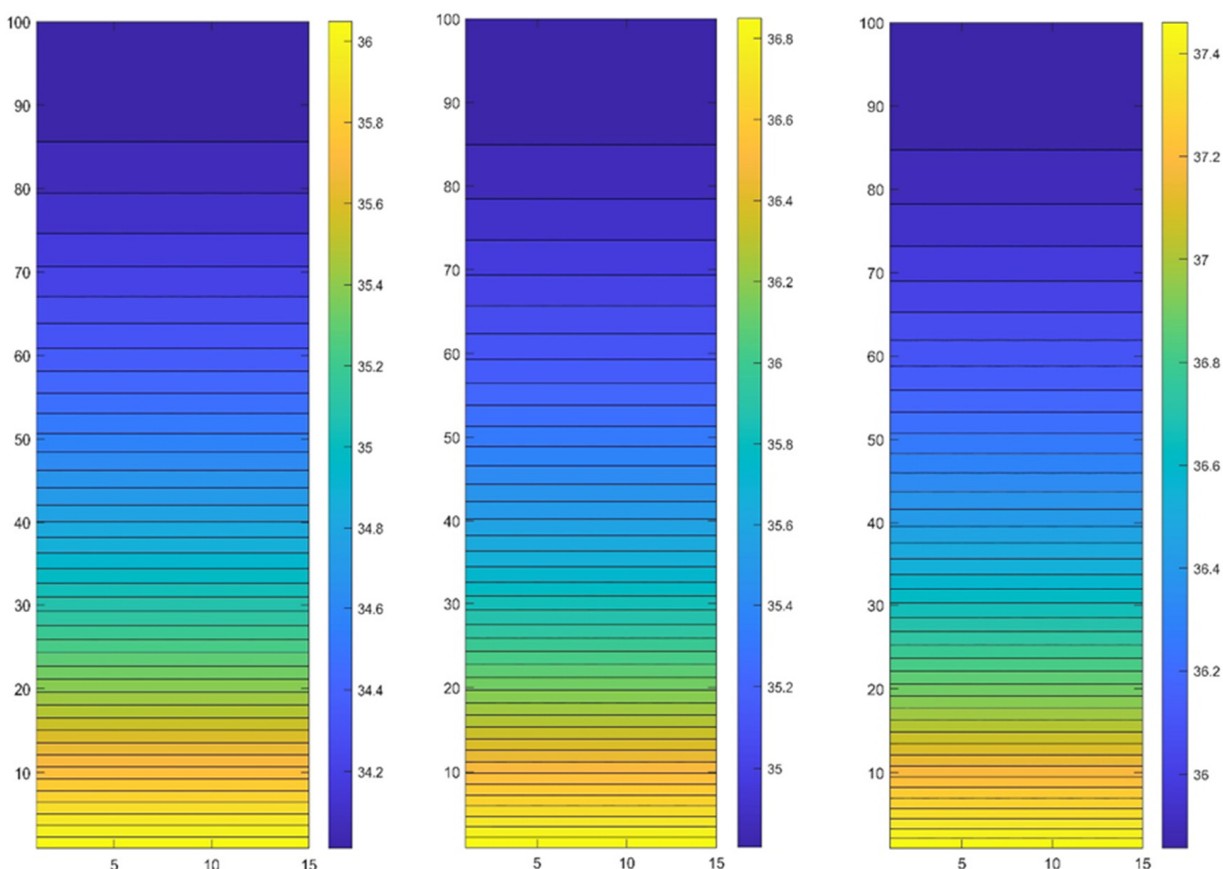

**Figure 11.** The final temperature distribution contour for different time values (t = 20, 24, and 30 min, respectively, from left to right) presented by the SH method when there is no heat exchange with the environment.

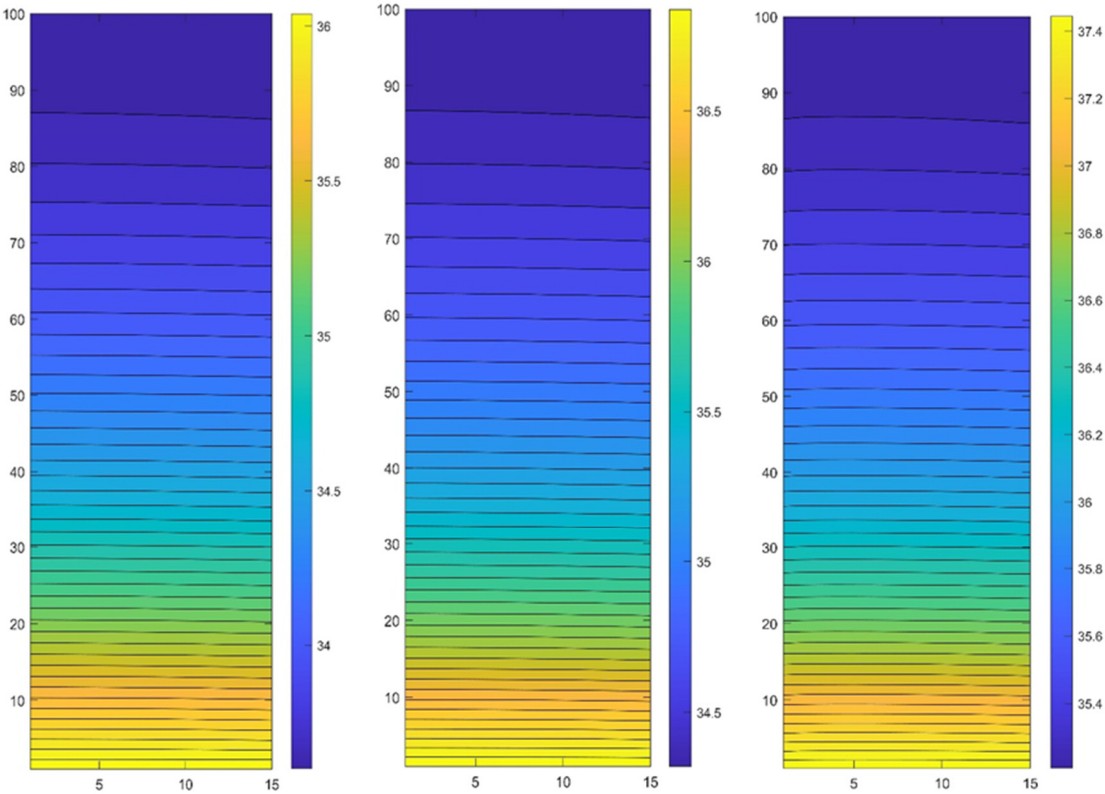

**Figure 12.** The temperature distribution contour for different time values (t = 20, 24, and 30 min) presented by the SH method when there is heat exchange with the environment via convection and radiation.

### 7.2. Ansys Simulation Results

Ansys workbench transient thermal analysis with Mechanical APDL solver was used to simulate the steel C45 cylinder. The mesh size was $1 \times 10^{-3}$, and the total number of elements was 197,183 since it was a computationally 3D problem. In Figures 13–15, we present the final temperature distribution at three measurement times.

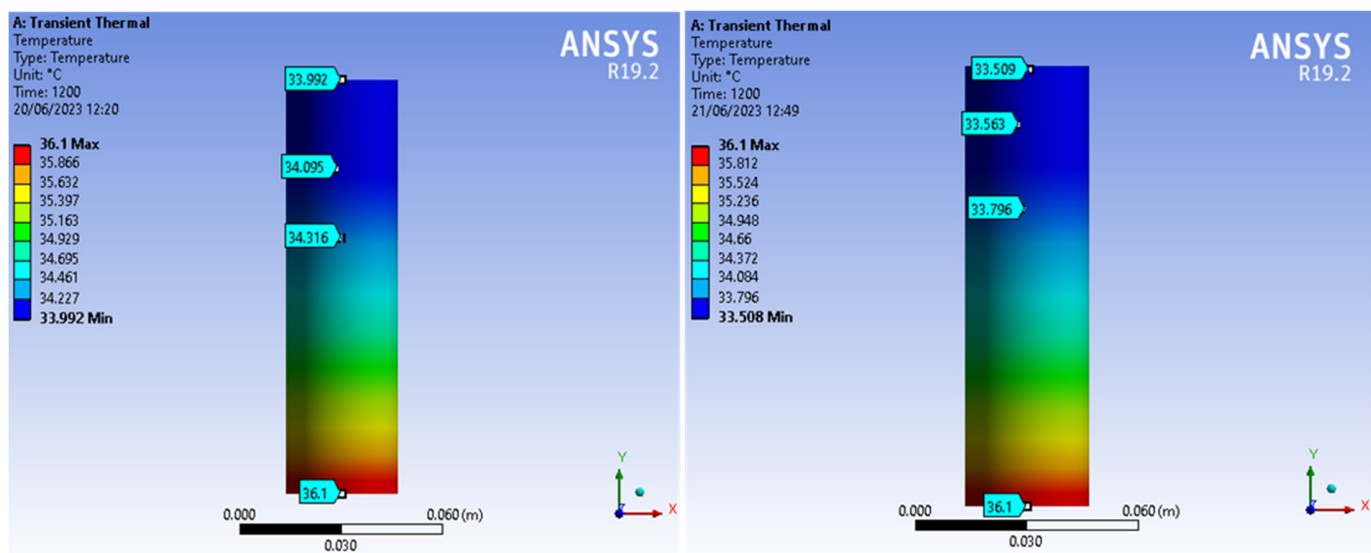

**Figure 13.** The temperature contour at time (t = 20 min) presented by Ansys workbench 19.2 when there is no heat exchange with the environment (**left**) and when there is a heat exchange (**right**).

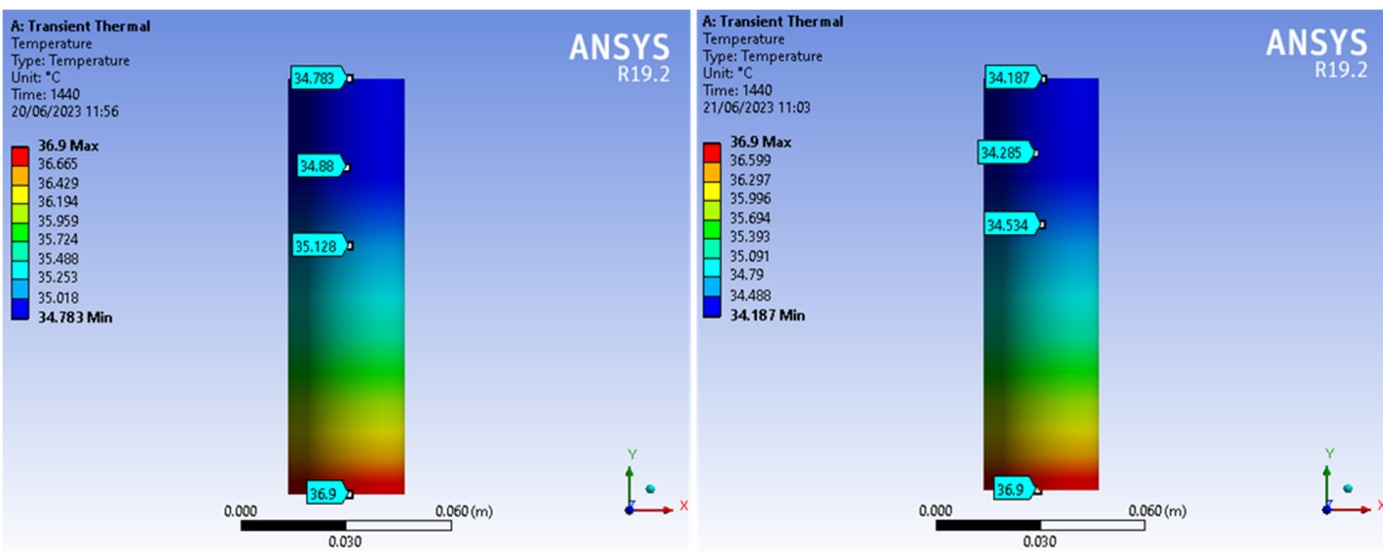

**Figure 14.** The temperature contour at time (*t* = 24 min) presented by Ansys workbench 19.2 when there is no heat exchange with the environment (**left**) and when there is a heat exchange (**right**).

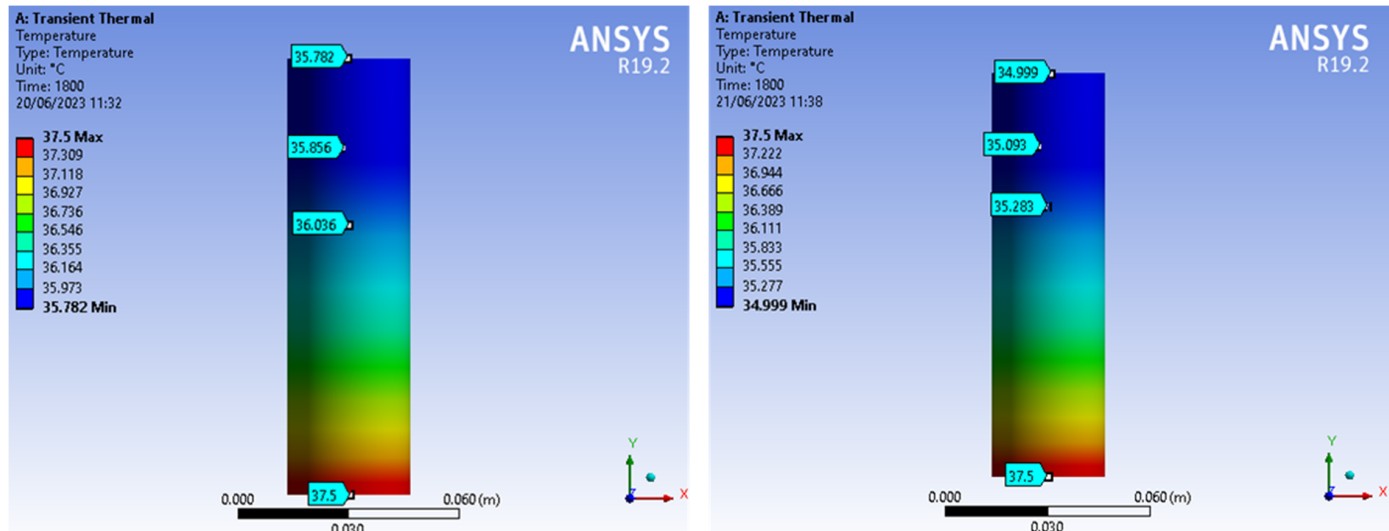

**Figure 15.** The temperature contour at time (t = 30 min) presented by Ansys workbench 19.2 when there is no heat exchange with the environment (**left**) and when there is a heat exchange (**right**).

*7.3. Comparison of the Results*

The results of the experimental measurements, the finite element method (FEM) using Ansys Workbench, and the explicit numerical methods (exemplified by the shifted hopscotch method) were compared. Both FEM and SH were subjected to two types of tests, one considering convection and radiation effects, and the other excluding them. First, we employed steady-state thermal analysis using FEM Ansys Workbench to follow the original paper [37] to reach the same results. The maximum deviation was 0.07, which was a kind of verification for setup. Then, we used transient thermal analysis to follow the real physical processes of the experiment. All results below are for this transient simulation. In Tables 2 and 3, the comparison was conducted at two specific spatial points (*z* = 75, and 95 mm, which are the distance from the bottom measurement point), and the results were measured at three different time moments. The temperatures are compared at two space points via plots in Figures 16–18.

**Table 2.** The temperature at $z = 125$ mm at three different time moments.

| Time | Temperature in °C, at $z = 75$ mm | | | | |
|------|------------|-----------|--------|-------------|--------|
|      | Experiment | SH with CR | SH    | FEM with CR | FEM    |
| 20 min | 33.9 | 33.941 | 34.298 | 33.796 | 34.316 |
| 24 min | 34.6 | 34.668 | 35.087 | 34.534 | 35.128 |
| 30 min | 35.7 | 35.514 | 36.07  | 35.283 | 36.036 |

**Table 3.** The temperature at $z = 145$ mm at three different time moments.

| Time | Temperature in °C, at $z = 95$ mm | | | | |
|------|------------|-----------|--------|-------------|--------|
|      | Experiment | SH with CR | SH    | FEM with CR | FEM    |
| 20 min | 33.7 | 33.71  | 34.099 | 33.563 | 34.095 |
| 24 min | 34.5 | 34.427 | 34.88  | 34.285 | 34.88  |
| 30 min | 35.5 | 35.30  | 35.92  | 35.093 | 35.856 |

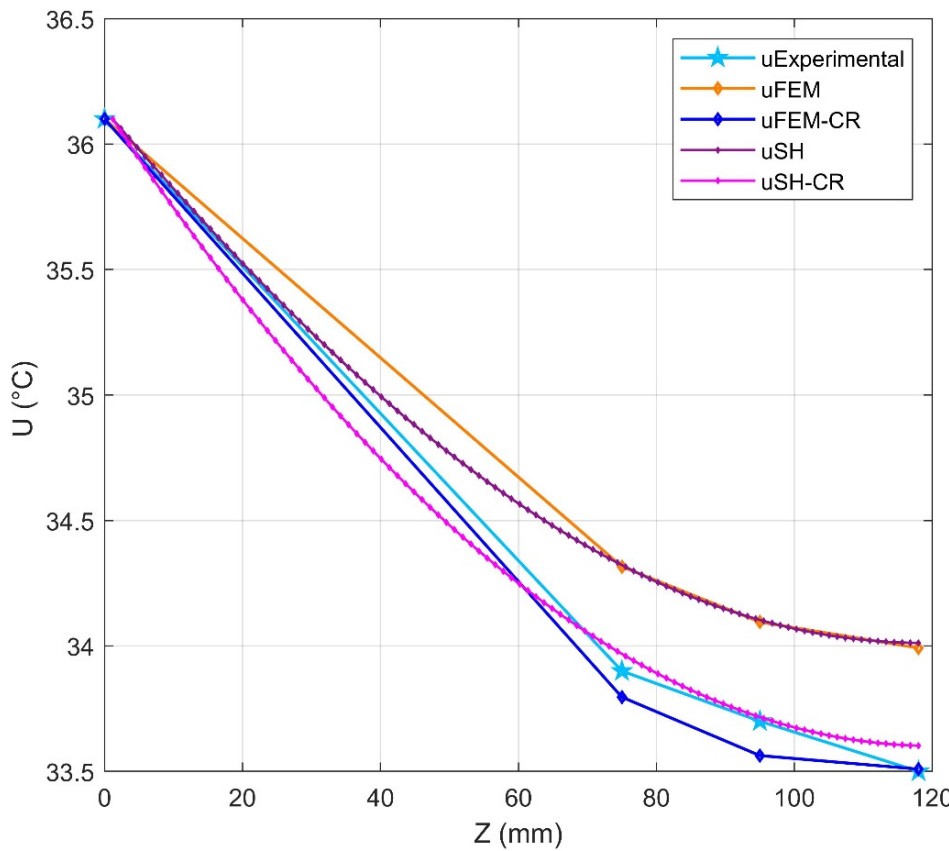

**Figure 16.** The temperature at the 4 selected measurement points in $z$ at time t = 20 min.

The figures and tables presented above illustrate a comparison of results obtained from the current numerical methods and the FEM ANSYS, utilizing experimental data from the literature study [37]. The findings indicate that the numerical methods employed in this study demonstrate superior accuracy compared with the FEM ANSYS used in both the current investigation and the same literature study [37].



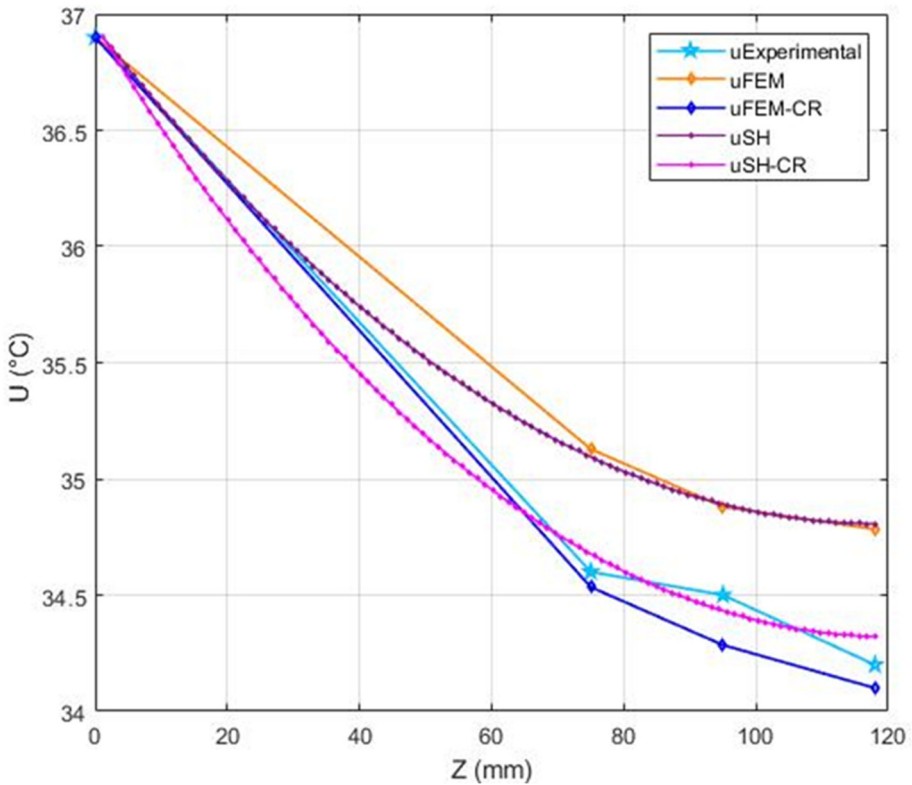

**Figure 17.** The temperature at the 4 selected measurement points in *z* at time t = 24 min.

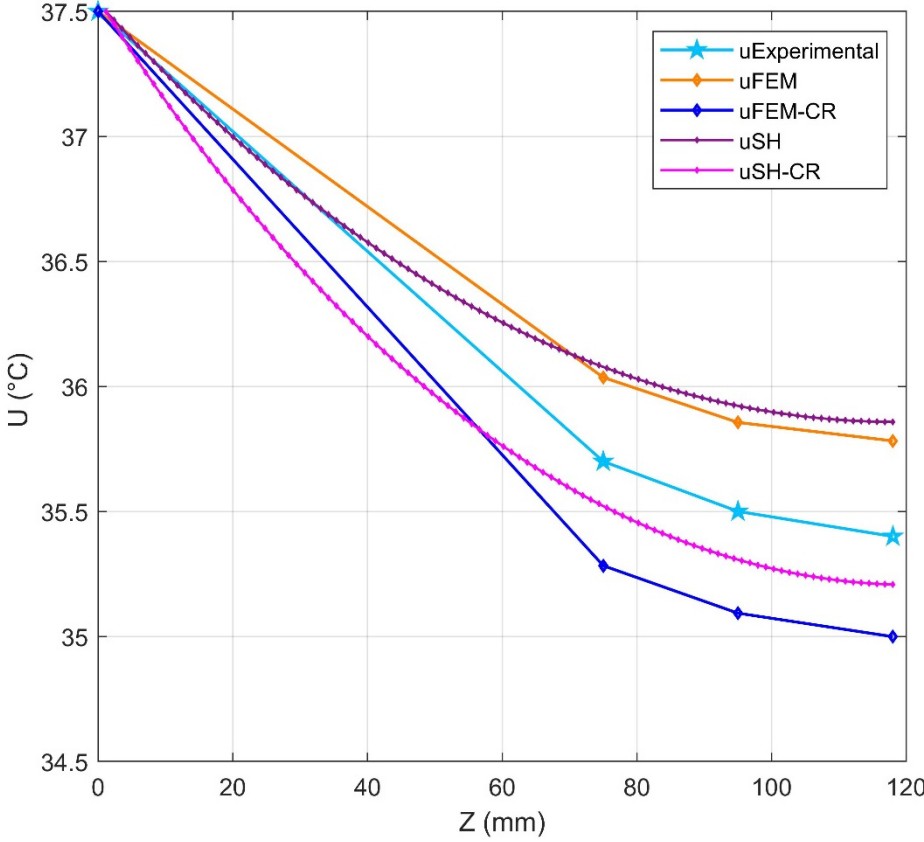

**Figure 18.** The temperature at the 4 selected measurement points in *z* at time t = 30 min.

## 8. Discussion and Summary

This work was devoted to solving heat transfer problems in cylindrical and spherical geometries. Using the self-similar Ansatz, novel analytical solutions of the heat-conduction PDE were constructed, which contained the Kummer's functions. Nine numerical algorithms were presented, most of which are recently introduced unconditionally stable explicit methods. To perform the verification, the analytical solutions were reproduced by these methods with high accuracy.

After these, experimental work was considered from the literature where a cylinder is heated from below, and the results were attempted to be reproduced using Ansys commercial software, but without considering convection and radiation on the surface of the cylinder. In contrast to that, we reproduced the experimental results by considering convection and radiation as well, not only using the Ansys, but the explicit methods as well. Since, in reality, convection and radiation are present, taking them into account makes the results closer to the experimental ones, especially for the first two measurement times. Moreover, the explicit and stable schemes were more accurate and effective than the finite element software in all cases. The LH algorithm was usually the most accurate among the studied methods. However, similarly to all hopscotch methods, it needs a special mesh, which can be hard or maybe impossible to implement for problems with irregular shapes. This limitation of these methods is probably more restrictive in complicated 3D problems.

**Author Contributions:** Conceptualization and methodology, E.K. and I.F.B.; supervision and resources, E.K.; analytical investigation and the related visualization, I.F.B. and L.M.; software, E.K. and H.K.J.; numerical investigation and the related visualization, H.K.J.; FEM investigation, H.K.J.; writing—original draft preparation, H.K.J.; writing—review and editing, E.K., H.K.J. and L.M. All authors have read and agreed to the published version of the manuscript.

**Funding:** This research received no external funding.

**Institutional Review Board Statement:** Not applicable.

**Informed Consent Statement:** Not applicable.

**Data Availability Statement:** Data are available on request from the first author.

**Conflicts of Interest:** The authors declare no conflict of interest.

## Nomenclature

| Quantity | Meaning, Unit |
| --- | --- |
| $u$ | Temperature, Kelvin (K) |
| $f$ | Shape function |
| $\eta$ | Reduced variable |
| $Q_{gen}$ | Heat Generation, Watt (W) |
| $Q_{convection}$ | Heat Convection, Watt (W) |
| $Q_{radiation}$ | Heat radiation, Watt (W) |
| $\Delta E$ | Change in Energy of an Element, Joule (J) |
| $k$ | Thermal Conductivity (W/(m·K) |
| $h$ | Convection Coefficient, (W/(m$^2$·K) |
| $\sigma*$ | Radiation Constant (W/(m$^2$·K$^4$) |
| $SB$ | Stefan–Boltzmann Constant W/(m$^2$·K$^4$) |
| $\alpha$ | (Thermal) Diffusivity |
| $\rho$ | Density (kg/m$^3$) |
| $c$ | Specific Heat (J/(kg·K) |
| $C$ | Heat Capacity (J/K) |
| $S$ | Surface Area (m$^2$) |
| $\Delta V$ | Element Volume (m$^3$) |
| $\varphi$ | Azimuthal angle, Rad, deg |
| $\theta$ | Polar angle, Rad, deg |
| $\Delta t$ | Time Interval (s) |

| | |
|---|---|
| *R* | Thermal Resistance (K/W) |
| *r* | Radius (m) |
| z | Height (m) |
| SH with CR | Shifted hopscotch with convection and radiation |
| FEM with CR | Finite element method with convection and radiation |

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
