# Peer review of "Analytical Solution and Numerical Simulation of Heat Transfer in Cylindrical- and Spherical-Shaped Bodies"

_computation, doi:10.3390/computation11070131_

Round 1
Reviewer 1 Report
In this work, the authors presented analytical solution as well as numerical simulation for heat conduction equation in cylindrical and spherical coordinates, and also compared their solutions with experimental data and with commercial software. Detailed comments are provided in the attached file.

In this work, the authors presented analytical solution as well as numerical simulation for heat conduction equation in cylindrical and spherical coordinates, and also compared their solutions with experimental data and with commercial software. Detailed comments are provided in the attached file.
Author Response
Response to Reviewer 1.
We are grateful to the reviewer for her/his precious time devoted to review our manuscript. Our point-by-point responses are the following.
- “Authors can provide some important results in the abstract as well as a brief outline of methodology.”
The abstract was modified according to the reviewer’s comment.
- “Grammatical errors are scattered throughout the manuscript and needs to be corrected.”
We have now carefully read the manuscript again and corrected all found grammatical errors.
- “A nomenclature seems to be important with different symbols and notations used in the study.”
We have now inserted a nomenclature table at the end of the paper.
- “The introduction needs to be revised. Firstly, the background of the current work is not clearly summarized. Also, how the current study is different from the existing one, particularly from the some of the references cited. The authors should clearly mention how their work is different from the existing studies and what improvement they made in their study? Also, identify the research gap between the current work and existing studies.”
The analytical solution (12) related to equation (7) is new, and we have never seen it in the scientific literature. We enhance this in section 3, immediately after the solution (12): “We consider these results original and novel, which are presented for the first time.”
Furthermore, this sentence defines the research gap from the numerical point of view: “However, all of these tests were performed in Cartesian coordinate systems, and now it is high time to perform them in cylindrical and spherical systems.”
- “The paper did not include the objective at the end of introduction. Also, explain the application background of the present paper in detail.”
In addition, we have also added additional explications related to the novelty of the results in Introduction.
We try to accentuate this already in the introduction as it was suggested by the reviewer.
In the last paragraph of the introduction we added:
“This means that the isotropic heat equation is solved for two and three dimensions in case of infinite horizon.”
We have now mention more applications in p. 1-2.
- “Authors mentioned that (in Section 2) they considered 2D assumption. In addition to this, they should also include axisymmetric assumption if they considered the same in their analysis. The paper did not include the objective at the end of introduction. Also, explain the application background of the present paper in detail.”
The three-dimensional spherical model which we consider and analyze is isotropic (the material constants and the temperature have the same property in all directions). The equations, which we solved, does not depend on the polar and azimuthal angles theta and phi.
Similarly, the cylindrical model is isotropic on all two dimensional projection planes which are parallel to the base of cylinder. There is no angle dependence.
Some applications are now mentioned.
- “What is . Authors need to define in the paper.”
is a general position vector. We agree with the reviewer that it can be confused with radial coordinate r, so we deleted from everywhere.
- “Line 114: Do you consider convection? Where are these effects shown in Eqs. (6&7).”
In Eq. (6), the term with coefficient h is responsible for the convection. In Eq. (7), as it is mentioned in the manuscript, this is taken as zero, so only conduction is considered.
Figures 3-10 for the conduction part only, while from Fig. 11 convection is also included.
- “In formulation of the problem, state all the necessary assumptions to derive the governing
equations and properly cite the respective references for the same.”
The necessary assumptions are listed in Eqs. (1)-(6). Now we have extended Eq. (6) with the notation “the density and the specific heat, respectively, which are considered as constants.”, thanks to the reviewer.
- “Do authors solve Eq. (7)? Please mention about the same.”
Yes, Eq (7) is solved analytically, the related sentences inserted to the manuscript are mentioned at points 4 and 5 above.
- ”In Line 122, check the definition of α.”
We have now specified that α is the “(thermal) diffusivity”.
- “Authors need to compare their solutions with proper validation with literature benchmark solutions.”
Most of our work was about validation/verification. We did more than it is usual: we did not take analytical solution from the literature, but constructed an own, completely new one. We also made a validation by comparison with the experimental data from the literature.
- “It appears that authors used Maple software to obtain the analytical solutions. Is there any methodology adopted to solve or they used built-in library to solve the model equation.”
It is important to emphasize that the derived solutions can be obtained with hand-and-paper methods as well with numerous tricky variable transformations one after another. So our ordinary differential equation (Eq. 8) can be transformed to the canonic form of the Kummer’s differential equation, which has the well-known analytical solution. Such solutions can be found in basic textbooks like NIST Handboook of Special Functions, or the good-old Abramowitz-Stegun handbook. (Such solutions can also be derived by the reader applying the Sommerfeld polynomial method, given in other textbooks.)
We have to note that there is no ‘built-in library’ in Maple to routinely solve these equations analytically. Some of the problems can be solved by Maple, while many others cannot, which is very hard to predict without trying it. The phrase “built-in library” is meaningful for numerical methods, such as the ODE solver Runge-Kutta, etc.
- “Also, explain in detail how this method is superior than the existing analytical solutions, such as variable separable method and Laplace transform method to solve similar equations.”
We do not necessarily consider one method superior to the other. Depending on the boundary conditions one method may be more appropriate than the other one. Variable separation and eventually Laplace transform is more adequate in case of finite horizon, while the self-similar Ansatz in case of infinite horizon. The latter may include the cases, where the system may be finite, but at least at one end there is no fixed value constraint for u. So we just present new and very interesting type of solutions with different kind of properties than traditional ones.
- “The conclusions must be More Specific of the obtained results. The present conclusions are more general. This section must be revised and authors need to state their key findings.
Also, setbacks or limitations of present work can also be stated.”
We have now revised the Conclusion section according to the comment of the reviewer. We also mention the limitations of the results in p. 5 and in the Conclusion section.
Reviewer 2 Report
This work's scope is focused on analytical and numerical solutions to the heat conduction equation presented in cylindrical and spherical coordinates. The subject of the research is interesting. I think this paper should be accept in present form.
Minor editing of English language required
Author Response
Response to Reviewer 2.
“This work's scope is focused on analytical and numerical solutions to the heat conduction equation presented in cylindrical and spherical coordinates. The subject of the research is interesting. I think this paper should be accept in present form.”
We are grateful to the reviewer for the very positive attitude towards our work. We have tried to do our best to further improve the manuscript.
Reviewer 3 Report
A review report is uploaded.

The authors should check the spelling and grammatical mistakes carefully
Author Response
Response to Reviewer 3.
We are grateful to the reviewer for her/his precious time devoted to review our manuscript. Our point-by-point responses are the following.
Major issues:
- “What are the practical applications of this study?”
We have now mentioned more applications in p. 1-2.
- “Did the authors check their results are sensitive due to the different timesteps and what are the CFL numbers?”
Figures 6, 7 and 9 about verification are devoted to this problem. They confirm our theoretical results that the explicit stable methods (LH, SH, ASH, OOEH, DF) are indeed unconditionally stable and the CFL limit (which is now calculated and given in Section 5, thanks to the reviewer) has no relevance for them. However, during the verification, we experienced that for larger time step sizes, the difference between the results of them are larger. On the other hand, when the results of these methods are very close to one another, all of them are very accurate. When we reproduced the experimental results, this latter case was achieved, and that is why we can state that the applied time step size in Section 7 is appropriately small.
- “Why the authors need the analytical solutions whereas anyone can get easily the numerical solution?”
Analytical solutions are always precious for the scientific community. One reason is that they help to verify the numerical methods. In general, they always help to better understand the general and global behaviour of the solutions, the role of the parameters, the asymptotic behaviour, etc. Our solutions are a kind of generalization of the old and well-known Gaussian solutions with more general properties, e.g. these are capable to describe oscillations as well. We hope that our solutions will attract attention in different fields e.g. statistics, quantum field-theory in the far future as well.
An analytic solution may show spatial distribution at a given time. In addition it may also show the rate of decay for long times. For example, this latter might be obtained numerically, but for a wide range of the spatial coordinates it means a huge amount of work.
- “Who did the experiment? No proper citation is here?”
We think it is clearly indicated, reference [41] in the current version.
- “In the figure 18, there are huge variation with experimental result. How can the authors justify that their results are good?”
We are grateful to the reviewer for this question. We have now repeat those simulations with an increased convection coefficient (based on Table 1-3 of the prestigious book [42]) and without fixing the temperatures of the top of the cylinder. Now we are quite close to the experimental results.
We mention that in the original experimental paper, the accuracy when they reproduced numerically their own experiment was worse.
“Minor issues:”
We have made all the corrections, thanks to the reviewer.
Round 2
Reviewer 1 Report
Authors have revised the paper in a satisfied manner. Hence, in my opinion, the revised paper has been accepted for publication.
Reviewer 3 Report
Recently, I have reviewed this manuscript and made some comments. The authors have revised the manuscript based on my observations.
Englsih is oaky.